# HyperTime: Hyperparameter Optimization for Combating Temporal Distribution Shifts

Shaokun Zhang
The Pennsylvania State University
State College, USA

Yiran Wu
The Pennsylvania State University
State College, USA

Zhonghua Zheng
The University of Manchester
Manchester, United Kingdom

Qingyun Wu
The Pennsylvania State University
State College, USA

Chi Wang
Microsoft Research
Redmond, USA

## ABSTRACT

In this work, we propose a hyperparameter optimization method named *HyperTime* to find hyperparameters robust to potential temporal distribution shifts in the unseen test data. Our work is motivated by an important observation that it is, in many cases, possible to achieve temporally robust predictive performance via hyperparameter optimization. Based on this observation, we leverage the 'worst-case-oriented' philosophy from the robust optimization literature to help find such robust hyperparameter configurations. HyperTime imposes a lexicographic priority order on average validation loss and worst-case validation loss over chronological validation sets. We perform a theoretical analysis on the upper bound of the expected test loss, which reveals the unique advantages of our approach. We also demonstrate the strong empirical performance of the proposed method on multiple machine learning tasks with temporal distribution shifts. The algorihtm is available in https://microsoft.github.io/FLAML/.

## CCS CONCEPTS

• **Computing methodologies** → **Artificial intelligence**; **Online learning settings**.

## KEYWORDS

Hyperparameter Optimization, Temporal Distribution Shift, Automatic Machine learning

**ACM Reference Format:**

Shaokun Zhang, Yiran Wu, Zhonghua Zheng, Qingyun Wu, and Chi Wang. 2024. HyperTime: Hyperparameter Optimization for Combating Temporal Distribution Shifts. In *Proceedings of the 32nd ACM International Conference on Multimedia (MM '24), October 28-November 1, 2024, Melbourne, VIC, Australia*Proceedings of the 32nd ACM International Conference on Multimedia (MM'24), October 28-November 1, 2024, Melbourne, Australia. ACM, New York, NY, USA, 13 pages. https://doi.org/10.1145/3664647.3681608

## 1 INTRODUCTION

One major hurdle for machine learning systems to effectively perform over time is *temporal distribution shifts*, which occur when the data distribution changes over time. If ignored, temporal distribution shifts may considerably degrade the predictive performance of the deployed machine learning models because of the data distribution mismatch during test time and train time [55]. In recent years, many methods have been proposed to improve ML model's robustness to distribution shifts in general, including continual learning [2, 10], invariant learning [3, 56], self-supervised learning [9, 11], and ensemble learning [25]. Although the methods mentioned above could potentially be adapted to handle temporal distribution shifts, the problem remains open and challenging: according to the evaluations from the Wild-Time benchmark [55], no existing invariant learning, continual learning, self-supervised learning, or ensemble learning approach is consistently more robust to temporal distribution shifts than vanilla empirical risk minimization (ERM).

In this work, instead of intervening in the ERM-based model training procedure [27, 59, 62], we approach the problem from a different perspective, hyperparameter optimization (HPO). It is known that some hyperparameters can affect the generalization capability [4, 48, 63, 64] of ML models. It is unknown, however, whether we can achieve temporally robust predictive performance via HPO. Figure 1 presents a case study on the Electricity dataset with temporal shifts. We observe that: (a) models trained based on different hyperparameter configurations may exhibit vastly different performances on chronologically out-of-sample test data, and (b) validation loss is positively correlated with test loss in general, but when the validation loss is close to the lowest, configurations with the same validation loss may still have significantly different test losses. The first observation indicates that it is possible to build ML models that are more robust to distribution shifts by performing hyperparameter tuning and model selection. The second observation suggests that it can be challenging to find such robust hyperparameter configurations.

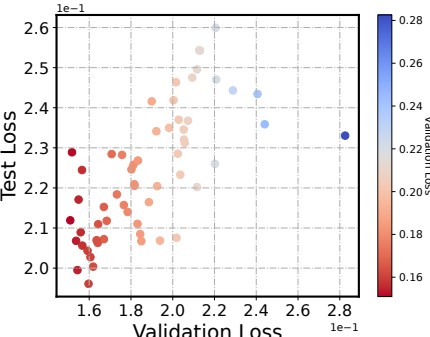

**Figure 1: Validation loss vs. test loss on the Electricity dataset, where the validation and test data are from different time periods. Each point is a hyperparameter configuration randomly sampled from the search space. The loss here is (1-ROC_AUC).**

In this work, we apply a principle from distributionally robust optimization [8, 12, 16, 46] to the regime of hyperparameter optimization. More specifically, when doing HPO in environments with temporal distribution shifts, instead of optimizing the average predictive performance on validation data, which are typically sampled uniformly at random, we propose to (1) construct validation sets from different time periods and treat them as different proxies for the unseen test data, and (2) consider both the average validation loss and worse-case validation loss during HPO. Specifically, we use a multi-objective HPO approach which allows a lexicographic structure [17] on the objectives to reflect the different priorities of the concerned objectives. We treat the commonly used average validation loss as the primary objective and the worst-case performance among the different subsets of the validation data as the secondary objective. This gives us the opportunity to leverage the worst-case performance toward finding robust configurations while respecting the importance of average validation loss. We provide theoretical analysis on the expected test loss of our method. The analysis shows the unique advantage of leveraging the average and worst-case validation loss in a lexicographic manner.

We verify the effectiveness of our method for tuning gradient-boosting trees and neural networks on a diverse range of datasets with temporal distribution shifts. Our method is also compatible with robust learning/training methods and is able to further boost their robustness to temporal distribution shifts.

## 2 RELATED WORK

A number of works are proposed to improve machine learning model's robustness when distribution shifts happen. One paradigm that can be applied is continual learning [2, 10, 20, 29, 44, 58] algorithms. The target of continual learning is to learn from new data on the fly while not forgetting previously learned information. Another paradigm that can be applied is invariant learning[18, 34, 47, 49, 54, 57]. Invariant learning methods aim to learn invariant representation across different domains, which could also be adapted to distribution shifts. The representative works include

CORAL [47], IRM [3], LISA [56], and GroupDRO [43]. Third, self-supervised learning [9, 11, 26, 45] and ensemble learning methods [15, 25, 41, 50] are also applicable to mitigating distribution shifts. All the aforementioned existing work concerns the training procedure to improve the resulting model's robustness. They are mostly model-specific and not consistently more robust to temporal distribution shifts than vanilla ERM according to the Wild-Time benchmark [55].

To the best of our knowledge, no existing HPO method concerns the temporal distribution shift problem. The only relevant work is a robust neural network search method named *NAS-OoD* [4], which searches for neural networks that generalize to out-of-distribution data under the differentiable neural architecture search paradigm [33]. However, this method is not model-agnostic and is not directly applicable to mitigate temporal distribution shifts.

## 3 METHOD

In this section, we present the proposed HPO method for combating temporal distribution shifts.

### 3.1 Notions, Notations, and Background

Before introducing details of the proposed method, we first introduce notions and notations to be used throughout the paper and some background knowledge on hyperparameter optimization and temporal distribution shifts.

(1) $(\mathbf{x}, y)$ denotes a specific supervised data instance where $\mathbf{x}$ represents the feature and $y$ represents the label. $\mathcal{D} = \{..., (\mathbf{x}, y), ...\}$ denotes a supervised dataset in general. When necessary, we use $\mathcal{D}_{t_1:t_2}$ to denote the subset of the dataset within a certain time period, e.g., $t_1$ to $t_2$.

(2) $c$ denotes a hyperparameter configuration in a particular hyperparameter search space $C$.

(3) $f$ denotes a machine learning model in general. When further details on the training data and hyperparameters are needed, we use sub-script $c$ and $\mathcal{D}$ in $f_{c,\mathcal{D}}$ to reflect that the model is constructed with a hyperparameter configuration $c$ and trained on dataset $\mathcal{D}$. We use $f(\mathbf{x})$ to denote the inference process on $\mathbf{x}$ outputting a predicted label.

(4) $\mathsf{Loss}(f, \mathcal{D})$ denotes the predictive loss of an ML model $f$ on dataset $\mathcal{D}$ under a particular loss function. For example, when Mean Squared Error is the loss function, we have $\mathsf{Loss}(f, \mathcal{D}) = \frac{1}{|\mathcal{D}|} \sum_{(\mathbf{x},y) \in \mathcal{D}} (f(\mathbf{x}) - y)^2$.

(5) We use $[K]$ as a shorthand for the set of integer from 1 to $K$, i.e., $[K] \coloneqq \{1, 2, ..., K\}$.

In a supervised machine learning setting, given a training dataset $\mathcal{D}_{\text{train}}$, the ultimate goal is to build a model $f$ based on $\mathcal{D}_{\text{train}}$ that has the best expected predictive performance on some unseen test data. Since the test data are unseen, a validation dataset is typically reserved (e.g., by sampling a particular portion uniformly at random) from the available training data as a proxy to evaluate

the predictive performance of the model on the unseen test data. In ML practice, validation loss is used ubiquitously as the primary metric for model selection in HPO and, more other machine learning tasks [37–39, 52, 53]. Specifically, a typical formulation of HPO is the following black-box optimization problem,

$$\min_{c \in C} \text{Loss}(f_{c, \mathcal{D}_{\text{train}}}, \mathcal{D}_{\text{val}}), \tag{1}$$

in which $\text{Loss}(f_{c, \mathcal{D}_{\text{train}}}, \mathcal{D}_{\text{val}})$ is the valuation loss on $\mathcal{D}_{\text{val}}$ corresponding to hyperparameter configuration $c$, and the objective of an HPO method under this formulation is to effectively find a hyperparameter configuration with the best validation loss. This optimization process is a principled approach for building different ML models [14, 23, 24, 61] in different tasks [13, 31, 32, 42] with good expected predictive performance on unseen test data when there is no distribution drift in the data (the expected predictive performance on the test data and validation data are supposed to be close according to theories in statistical machine learning [1]). However, when there is indeed data distribution drift, the optimization objective specified in Eq. (1) becomes questionable because of the mismatch between the predictive performance on the validation and test data due to distribution shifts.

## 3.2 Robust HPO by Imposing Lexicographic Objectives

Our overarching insight for doing robust HPO is to construct a set of possible realizations of the unseen test data and take the worst-case realizations into consideration in the hyperparameter optimization objectives.

To implement this idea, we first construct $K$ validation sets, denoted by $\{\mathcal{D}_1, \mathcal{D}_2, ..., \mathcal{D}_K\}$, which are possible realizations of the unseen test data. Based on the $K$ validation sets, we could obtain a set of validation losses denoted by $\{L_1(c), L_2(c), ..., L_K(c)\}$ respectively. We further denote the average loss and the worst loss among the $K$ losses as,

$$L_{\text{avg}}(c) := \frac{\sum_{i=1}^{K} L_k(c)}{K}, L_{\text{worst}}(c) := \max\{L_k(c)\}_{k \in [K]}. \tag{2}$$

If the data distribution in the unseen test set follows the same distribution as in the validation data, optimizing the average loss $L_{\text{avg}}(c)$ is presumably a good practice, which is also the standard practice in classical HPO when cross-validation is used. However, in the scenarios where temporal shifts exist, this assumption is no longer true, and better practice is needed. Inspired by the "worst-case-oriented" philosophy in robust optimization [8, 12], we propose to incorporate the validation loss on the fold with the worst predictive performance, i.e., $L_{\text{worst}}(c)$, as an additional objective for HPO.

**Lexicographic Hyperparameter Optimization.** It remains a question how one should incorporate the worst-case performance into consideration, especially regarding its relationship with average performance. In this work, we propose to include both average validation loss and worst-case validation loss during HPO and impose a lexicographic priority order on them. More specifically, we

include the ordered list $\mathbf{L}(c) = [L_{\text{avg}}(c), L_{\text{worst}}(c)]$ as objectives with a lexicographic structure, in which $L_{\text{avg}}(c)$ is the objective with higher priority and $L_{\text{worst}}(c)$ as the one with lower priority. By doing so, we could find a hyperparameter configuration with both a good average validation loss and a good worst-case validation loss over the validation folds. Put more formally, we formulate the HPO process as:

$$\text{LexiMin}_{c \in C} \mathbf{L}(c), \tag{3}$$

in which LexiMin is the optimization procedure over an ordered list of objectives $\mathbf{L}(c)$, following the Lexicographic relations defined in [60]. We use $L^{(i)}$ to denote the $i$-th element of the list $\mathbf{L}(c)$ in general. In our optimization function, $L^{(1)}$ and $L^{(2)}$ represents $L_{\text{avg}}$ and $L_{\text{worst}}$, respectively. Given any configurations $c$, $c'$, and $I = |\mathbf{L}(c)|$ (with $I > 1$) optimization objectives with a lexicographic priority order, the definition of lexicographic relation (between any $c' \in C$ and $c \in C$) is:

$$\mathbf{L}(c') =_l \mathbf{L}(c) \iff \forall i \in [I] : L^{(i)}(c') = L^{(i)}(c), \tag{4}$$

$$\mathbf{L}(c') \prec_l \mathbf{L}(c) \iff:$$
$$\exists i \in [I] L^{(i)}(c') < L^{(i)}(c) \land (\forall i' < i, L^{(i')}(c') = L^{(i')}(c)),$$
$$\mathbf{L}(c') \preceq_l \mathbf{L}(c) \iff \mathbf{L}(c') \prec_l \mathbf{L}(c) \lor \mathbf{L}(c') =_l \mathbf{L}(c).$$

The optimal point under LexiMin is called the *lexi-optimal* point, which is any one element in hyperparameter configuration set $C^* = \{c \in C_*^{(I)} | \forall c' \neq c, \mathbf{L}(c) \preceq_l \mathbf{L}(c')\}$. Here $C_*^I$ is defined in the following recursive way: $C_*^{(0)} = C$ and for $i \in [I]$,

$$C_*^{(i)} := \{c \in C_*^{(i-1)} | L^{(i)}(\mathbf{x}) \leq L_*^{(i)} * (1 + \kappa^{(i)})\}, \tag{5}$$

$$L_*^{(i)} := \inf_{c \in C_*^{i-1}} L^{(i)}(c),$$

where $\kappa^{(i)}$ is a non-negative number, representing the percentage of performance compromise of the $i$-th objective to find choices with better performance on the low-priority objectives.

Compared with directly using the average validation loss $L_{\text{avg}}$ as the single optimization objective, LexiMin is able to incorporate an auxiliary objective $L_{\text{worst}}$ by adding it as the secondary objective in lexicographic preference. In this way, the optimization of $L_{\text{worst}}$ only matters when the more important objective $L_{\text{avg}}$ is well-optimized, i.e., within its optimality tolerance range. Compared to classical multi-objective HPO approaches, LexiMin is able to incorporate the intuition that the average loss shall be prioritized. We modify the HPO solution designed for this type of LexiMin problem originally proposed in [60] to solve our problem after constructing the objectives. We include the algorithm details in the Appendix A.

*Remarks on validation data sets construction.* In addition to the lexicographic objectives on the average validation loss and the worst-case validation loss, we believe it is also important to consider how the validation shall be constructed.

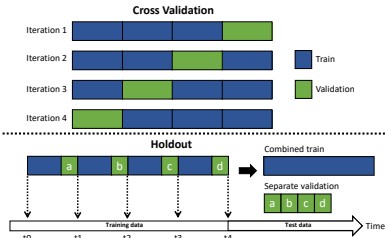

**Figure 2: Chronological validation data sets construction with Cross Validation and Holdout strategies.**

The principle for constructing the validation sets is that the validation sets should represent possible realizations of unseen data. Considering this and the potential temporal distribution shifts in the dataset, we propose to retain the chronological order over the data instances and sample the $K$ folds of validation data $\mathcal{D}_1, \mathcal{D}_2, ..., \mathcal{D}_K$ at different time periods. Specifically, we first split the chronologically ordered training dataset into $K$ segments with $K-1$ time points $t_1, ..., t_{K-1}$ in addition to the starting point $t_0$ and the end point $t_K$ (the actual value of the time points can be application dependent). We then ensure $\forall k \in [K]$ the validation set $\mathcal{D}_k$ is sampled from time period between $t_{k-1}$ to $t_k$. By doing so we have a collection of diverse validation sets representative of the potentially shifted data distributions in the available training set.

Depending on whether cross-validation or holdout is preferred, the validation set construction strategy and the corresponding calculation of validation losses in both typical cross-validation and holdout are visualized in Figure 2 and detailed formally as follows, in which we use $\mathcal{D}$ to denote the available dataset: **(1) Cross-validation:** Each evaluation of a particular configuration $c$ involves $K$ iterations of model training and evaluation. In the $i$-th iteration, the set $\mathcal{D}_i = \mathcal{D}_{t_{k-1}:t_k}$ is considered the validation set and the rest training set. And we have $L_i(c) := \text{Loss}(f_{c,(\mathcal{D}\setminus\mathcal{D}_i)}, \mathcal{D}_i)$ for $i \in [K]$. **(2) Holdout:** In this case, the evaluation of each configuration only involves training one single model with $K$ validation steps. The $k$-th validation set is $\mathcal{D}_i = \mathcal{D}_{t'_{k-1}:t_k}$ in which $t_{k-1} < t'_{k-1} < t_k$, and the data excluding the $K$-folds of valiation sets, i.e., $\mathcal{D} \setminus (\mathcal{D}_1 + \mathcal{D}_2 + ... + \mathcal{D}_K)$, are used to train a model. And we have $L_i(c) := \text{Loss}(f_{c,(\mathcal{D}\setminus(\mathcal{D}_1+\mathcal{D}_2+...+\mathcal{D}_K))}, \mathcal{D}_i)$ for $i \in [K]$.

Although cross-validation is usually the preferred method because it allows models to train on multiple train-test splits, each evaluation of a particular configuration is more expensive than the holdout strategy (approximately $K$ times larger), especially in hyperparameter tuning which depends on a large number of configuration evaluation processes. Therefore, we suggest choosing the validation sets construction method according to the detailed information of the scenario like data size, model types, resource limit, etc.

## 4 THEORETICAL ANALYSIS

In this section, we attempt to provide a theoretical analysis of the proposed hyperparameter optimization algorithm. The main objective is to provide an upper bound for the expected test loss of

the model with the selected hyperparameter of our method. As an initial attempt, the analysis requires some assumptions. We leave the relaxation of these assumptions to future work.

Following the same spirit as previous works on mitigating distribution shifts occurred with time in data stream [21, 36], we assume that among the (K) validation sets from previous time periods, the optimal configuration is the one that performs best on one particular validation sets that shares *the most similar* data distribution with the unseen test data at recent time periods. This assumption is an important relaxation of the full i.i.d. assumption required by existing HPO algorithms [7, 16]. We further introduce the following definitions to facilitate our analysis.

(1) Best configuration on the $k$-th validation data set $c_k^*$: $c_k^* := \arg\min_{c \in C} \text{Loss}(f_{c,\mathcal{D}_{\text{train}}}, \mathcal{D}_k)$.

(2) Best average validation loss $L_{\text{avg}}^*$: $L_{\text{avg}}^* := \arg\min_{c \in C} L_{\text{avg}}(c)$.

(3) We use $k^*$ to denote the index of the validation set that shares the most similar data distribution with the unseen test data. In other words, validation set $D_{k^*}$ shares the most similar data distribution with $D_{\text{test}}$.

(4) We use $\hat{c}$ to denote the hyperparameter selected by our method.

As defined above, the configuration $c_{k^*}^*$ is the optimal configuration. However, $k^*$ is unknown a prior without the test data. We provide the following bound on the validation loss of our selected configuration $\hat{c}$ and a proof sketch as well as the detailed proof in Appendix B for Theorem 2.

LEMMA 1. *When $\mathcal{D}_{test}$ and one particular $\mathcal{D}_{val}$ are from the same distribution, then for any $c \in C$, with probability at least $1 - \epsilon$ ($\epsilon \in (0, 1)$), we have:*

$$|\text{Loss}(f_c, \mathcal{D}_{val}) - \mathbb{E}[\text{Loss}(f_c, \mathcal{D}_{test})]| \leq \sqrt{\frac{\beta \ln(1/\epsilon)}{2|\mathcal{D}_{val}|}},$$

*in which $\beta$ is the distance between the largest and the lowest loss value on any data instance.*

Assuming there exists one particular validation data $\mathcal{D}_{\text{val}}$ (not necessarily exists) that shares the same distribution with the unseen test data $\mathcal{D}_{\text{test}}$, then Lemma 1 provides an upper bound for distance between the expected test loss and the validation loss. The proof for Lemma 1 could be found in Appendix B.

THEOREM 2. *When $\kappa \geq \frac{L_{avg}(c_{k^*}^*)}{L_{avg}^*} - 1$, with probability at least $1 - \epsilon$ ($\epsilon \in (0, 1)$), we have the following bounds on the expected test loss of the model with our selected configuration $\hat{c}$,*

$$\mathbb{E}[\text{Loss}(f_{\hat{c}}, \mathcal{D}_{test})] \leq \begin{cases} if\ L_{k^*}(\hat{c}) \leq L_{avg}(\hat{c}): \\ (1+\kappa)L_{avg}(c_{k^*}^*) + \sqrt{\frac{\beta \ln(2/\epsilon)}{2|\mathcal{D}_{val}|}}, \\ Otherwise: \\ L_{worst}(c_{k^*}^*) + \sqrt{\frac{\beta \ln(2/\epsilon)}{2|\mathcal{D}_{val}|}}, \end{cases}$$

*in which $\beta$ is the upper bound on the loss. E.g., in binary classification task with 1-accuracy as the loss metric, $\beta = 1$.*

**Table 1: Test time performance of HyperTime and baselines for tuning gradient-boosting trees on different datasets. We show the average test loss (Test-average), and average worst fold test loss (Test-worst) across test folds with 5 seeds respectively. The losses are the lower the better. For each method, we also show the number of folds achieving the best results compared with other methods, i.e., winning fold num (WN), which is the higher the better. Considering the loss numerical values are not understandable, we normalize the loss values for each evaluation metric across all methods using Z-score normalization[1].**

| Metric | Vessel Power | | | Temperature | | | Electricity | | |
|---|---|---|---|---|---|---|---|---|---|
| | Test-average | Test-worst | WN | Test-average | Test-worst | WN | Test-average | Test-worst | WN |
| Default | 2.028 | 1.425 | 2 | 5.606 | 5.392 | 0 | 1.359 | 2.554 | 2 |
| CFO | 2.770 | 2.217 | 0 | 3.692 | 3.738 | 0 | 3.007 | 3.717 | 0 |
| BO | 4.316 | 3.108 | 0 | 2.991 | 2.785 | 0 | 2.263 | 2.276 | 0 |
| HB | 3.656 | 3.098 | 0 | 3.628 | 3.505 | 0 | 3.027 | 4.339 | 0 |
| HyperTime | **1.616** | **0.511** | **5** | **2.768** | **2.552** | **7** | **0.434** | **1.576** | **4** |

*Remark* 4.1 (The role of $\kappa$). According to the analysis in Appendix B, we have:

**(1)** When $L_{k^*}(\hat{c}) \leq L_{avg}(\hat{c})$, a smaller $\kappa$ shall be preferred. In fact, under this case, if we set $\kappa$ to 0, and the method recovers the naive alternative, which uses the average validation loss as the HPO objective.

**(2)** When $L_{k^*}(\hat{c}) > L_{avg}(\hat{c})$, using the average validation loss is no longer a good strategy as it may make the expected test loss $\mathbb{E}[\text{Loss}(f_{\hat{c}}, \mathcal{D}_{\text{test}})]$ as large as $KL_{\text{worst}}(c_{k^*}^*) + \sqrt{\frac{\beta \ln(2/\epsilon)}{2|\mathcal{D}_{\text{val}}|}}$. With our method, as long as $\kappa$ satisfies $\kappa \geq \frac{L_{avg}(c_{k^*}^*)}{L_{avg}^*} - 1$, $\mathbb{E}[\text{Loss}(f_{\hat{c}}, \mathcal{D}_{\text{test}})]$ is upper bounded by $L_{\text{worst}}(c_{k^*}^*) + \sqrt{\frac{\beta \ln(2/\epsilon)}{2|\mathcal{D}_{\text{val}}|}}$ with high probability.

Considering the fact that $k^*$ is unknown a prior (in other words, which fold of the validation data is most similar with the test data is unknown a prior), both case (I) and case (II) may happen. Our method is able to properly bound the expected test loss in both cases despite the value of $k^*$ is unknown.

## 5 EXPERIMENTS

We begin by providing the datasets and corresponding experimental setting in Section 5.1. We then evaluate our method (*HyperTime*) on the gradient-boosting trees and neural networks tuning tasks in Section 5.2 to verify the effectiveness of our method. We further perform in-depth investigations in Section 5.3 to (1) provide a better understanding of the important contributing factors in our method; and (2) study the compatibility of our method with robust training methods. If not otherwise specified, all the results in our evaluation are averaged over five different random seeds. .

### 5.1 Datasets

Before the experiments, we first introduce the main datasets we employed in tuning both gradient-boosting trees and neural networks.
(1) Electricity: A classification task. It is widely used for evaluating distribution shifts mitigation methods [36]. The dataset contains two and a half years of data. We exclude the first half

year and use the next one year for training and the last year for testing. We split every 2 months into one fold.
(2) Vessel power estimation: A regression task taken from Wild-Time benchmark [35]. It is a large dataset with 523,190 training samples over 4 years, and we use the out-of-distribution dev-set as our test data which has 18,108 samples. We split the training data uniformly into 12 folds, and the test data into 7 folds.
(3) Urban temperature prediction: A regression task to predict the urban daily maximum of average 2-m temperature. It has distribution shifts as mentioned in [28, 40]. We split every 5 years into one fold and we use the first 40 years for training and test on the remaining 35 years.
(4) YearBook: Yearbook is an image dataset with 37,921 frontal-facing American high school yearbook photos from 1930 - 2013. Each data point is a $32 \times 32 \times 1$ grey-scale image and the label is the student's gender. Distribution shifts occur due to social norms, fashion styles, and population demographics changing over time. Following the same setting with Wild-Time, we use 1970 as the split timestep to split the training and test set.

### 5.2 Effectiveness

In this subsection, we show the off-the-shelf effectiveness of our proposed method for tuning tree-based boosting methods and deep neural networks. We include three single objective HPO methods as baselines in all the evaluations, including randomized direct search method [51] (CFO), bayesian optimization HPO algorithm [6] (BO), and multiple multi-fidelity HPO algorithm [30] (HB), which search for the best configuration that maximizes the average validation losses. In the task of boosting trees tuning, we also include the learners with default configuration, as baselines. This baseline can be considered as an ERM method under the tree-based boosting framework. In the task of deep neural network tuning, we include state-of-the-art robust training methods (including a vanilla ERM as well) for comparison. The detailed search spaces for each learner are included in Appendix C.

We use three metrics to perform evaluations on the test set, which could reveal the test performance of a method from multiple aspects. **(1) Average performance:** Average performance of all test folds. It reflects the overall performance of a specific method, and it is typically considered the most important metric in practice. **(2) Worst fold performance:** Worst fold performance across all test folds. It reflects the performance of a specific method in the worst cases. **(3) Winning fold number:** Number of test folds achieving

---

[1]To ensure that the normalized values remain positive, we introduce a shift equal to twice the maximum value of the normalized results across all methods.

the best performance compared with other methods. When temporal distribution shift happens, assuming each test fold follows one specific data distribution, winning fold number could reflect the number of cases in which a specific method works best compared with other methods.

*5.2.1 Tuning tree-based boosting methods.* We first perform the evaluation for tuning different gradient-boosting trees on three tabular datasets, including two large-scale datasets Vessel Power Estimation [35] and Urban Temperature Prediction, and a relatively small dataset Electricity [36] to cover a wide use cases. We tune XGBoost on the Electricity and Vessel Power Estimation datasets, and LightGBM on the Urban Temperature Prediction dataset [28]. Note that in CFO, we use the conventionally used validation data set construction, i.e., constructing validation sets by randomly sampling from shuffled datasets. We report the normalized average test loss, normalized worst fold test loss, and the winning fold number in Table 1. Compared with all baselines, HyperTime achieves the best performance in terms of both average performance and the worst fold performance on all three datasets. It indicates our method could indeed help find hyperparameter configurations with relatively robust performance during test time.

We also present the predictive performance on each fold of the test data in Figure 3. Figure 3 shows that HyperTime is consistently better than the baseline methods on different test folds in most cases. Although there are cases where the baseline methods have better performance than HyperTime on a specific fold, the margin of the differences is small.

We also have an interesting observation: Vanilla HPO (CFO) with the average validation loss as the objective is worse than the default learner in two of three datasets (2/3). This scenario also appears in pioneer works [5] and it reflects the motivation of our paper to some extent. Single-objective HPO algorithms only use the validation loss as the optimization objective, which may cause the searched architectures to overfit the validation data. This overfitting scenario in HPO has also been justified in [60].

*5.2.2 Tuning neural networks.* We perform a neural network tuning task on a large image classification dataset Yearbook from the Wild-Time benchmark [55], which consists of 33,431 American high school yearbook photos. Due to the change of social norms, and other potential factors that may change with the passage of time, there exist temporal distribution shifts in it [19].

To make our evaluation more comprehensive and convincing, in addition to the single-objective HPO baselines, we also include the state-of-the-art robust training methods that are applicable to this task. For each type of method mentioned in Wild-Time, we choose one algorithm with the best average test performance according to the benchmarked results. Specifically, we include the classic supervised learning method ERM, a continual learning method Fine-tuning, temporal invariant learning method LISA [56], a contrastive learning method SimCLR [11] and a Bayesian learning method SWA [25]. We use the implementations for those methods from Wild-Time and follow the same Eval-Fix evaluation setting with the benchmark.

Table 2 shows the final test results from HyperTime and all the compared methods. In terms of average performance and the worst fold performance, we observe that HyperTime is the best one compared to others. Moreover, we also observe that the performance of the HPO algorithms (single HPO algorithms and HyperTime) are significantly better than the non-HPO methods. We also show the winning number for each method in Table 2, HyperTime gets the best results on 7/9 of the test folds which is significantly better than other methods. In summary, the effectiveness of HyperTime is evidenced by its superior performance compared to single objective HPO algorithms such as CFO, BO, and HB, as well as other state-of-the-art non-HPO methods across various tasks.

*5.2.3 HyperTime is consistently better than ERM on Wild-Time [55].* One of the key findings from the Wild-Time benchmark [55] is that no existing method consistently outperforms Empirical Risk Minimization (ERM) across all datasets in the Wild-Time benchmark. This observation underscores the strength of ERM as a strong baseline for addressing temporal distribution shifts.

Motivated by this insight, we conducted comprehensive experiments to compare HyperTime with ERM. Specifically, we rigorously adhered to the same Eval-Fix setting used in Wild-Time [55] and adapted HyperTime to fit this evaluation framework. We show the Hyperparameter we used in the experiments in Table 9 of Appendix C.2. Our results, presented in Table 4, demonstrating that HyperTime consistently outperforms ERM on all datasets within the Wild-Time benchmark. These findings highlight the promise of HyperTime as a method that surpasses existing approaches.

## 5.3 Further Investigation

In this subsection, we conduct further investigations for our method including ablation studies and an evaluation of our method when combined with robust training methods.

*5.3.1 Ablation.* We first do a series of ablation studies aiming to provide a better understanding regarding the two important components of our method: (1) Regarding the validation sets: Does the chronological re-sampling strategy matter when constructing the validation sets in our method? (2) Regarding the optimization objectives: Are there easy alternatives to achieve similarly good performance?

*The construction of validation sets.* We first perform experiments to investigate the validation sets construction part in HyperTime. We construct the following two variants of CFO and HyperTime by changing the way the validation sets are constructed to study how these changes impact the final performance: **(1) HyperTime-w/o-chronology:** In this method, we do not use the chronologically constructed validation sets and instead construct validation sets by randomly re-sampling from shuffled datasets (no-chronological-order, conventional approach in practice). **(2) CFO-w/-chronology:** In this method, we add the chronologically constructed validation sets in the standard CFO.

We compare the performance of (1) and (2) with their original versions, i.e., CFO and HyperTime on the Electricity dataset and

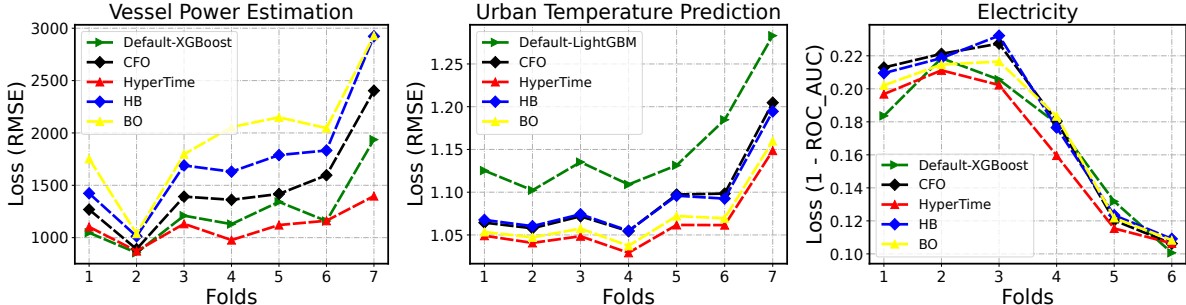

**Figure 3: Per fold test loss (lower the better) for tuning gradient-boosting trees on different datasets. The results are averaged over different random seeds. The results are from the same set of experiments with that in Table 1.**

**Table 2: The results of baselines and our method on the yearbook dataset. We show the average test accuracy, the worst fold accuracy, and the number of winning folds (WN) across 9 test folds with 3 seeds, which are denoted as Test-average, Test-worst, and Winning fold num, respectively. All the numbers are the higher the better.**

|  | ERM | Fine-tuning | LISA | SIM-CLR | SWA | CFO | BO | HB | HyperTime |
|---|---|---|---|---|---|---|---|---|---|
| Test-average | 77.74 | 79.09 | 83.45 | 74.72 | 82.60 | 83.88 | 83.55 | 83.83 | **84.58** |
| Test-worst | 65.24 | 70.09 | 70.74 | 62.69 | 71.57 | 73.05 | 71.23 | 70.43 | **73.91** |
| WN | 0 | 0 | 2 | 0 | 0 | 0 | 0 | 0 | **7** |

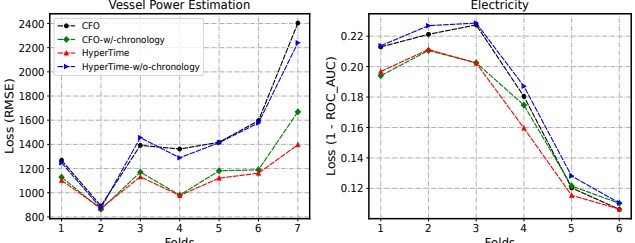

**Figure 4: Test loss of CFO and HyperTime on different folds with/without using chronological validation sets.**

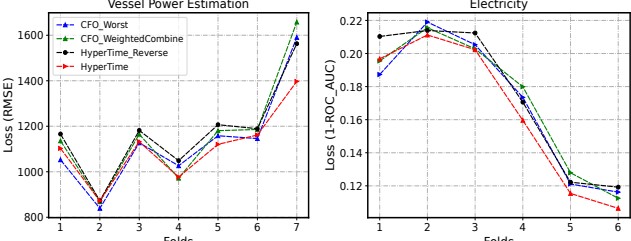

**Figure 5: Test loss of different folds using HyperTime, Hyper-Time_Reverse, CFO_WeightedCombine and CFO_Worst.**

**Table 3: The feature number, instance number, and valida-tion/test folds number of each dataset in the paper.**

|  | Feat. num | Inst. num | Val. num | Test num |
|---|---|---|---|---|
| **Electricity** | 8 | 33873 | 6 | 6 |
| **Vessel. estimation** | 11 | 541298 | 12 | 7 |
| **Temp. prediction** | 10 | 437884 | 8 | 7 |
| **YearBook** | \ | 33431 | 8 | 9 |

Vessel power dataset. We show the test results of these methods in Table 5 and we also include the test results of different folds in Figure 4. We observe that the methods with chronological validation sets (CFO-w/-chronology and HyperTime) are obviously better than their corresponding versions with random validation sets (CFO and HyperTime-w/o-chronology). This indicates that chronological cross-validation is indeed an important contributing factor to the good performance of HyperTime.

*Optimization objectives.* We then perform experiments to investigate the role lexicographic optimization plays in our method. We vary optimization objective formulations in our method in different ways and investigate the factors in the objective formulations that make contributions to the final performance. We construct three new methods for comparison as shown below:

**(1) CFO_Worst:** Using chronological validation sets and setting the worst-fold validation loss as the objective in CFO. **(2) Hyper-Time_Reverse:** Reversing the priority of optimization objectives in our method, i.e., setting the worst-fold validation loss as the primary objective and the average validation across folds as the secondary objective. **(3) CFO_WeightedCombine:** Using chronological validation sets and setting the optimization objective as a weighted combination of two objectives in CFO. Weights are 0.99 and 0.01 for average validation loss and the worst fold validation loss, respectively, which is consistent with the tolerance setting in our experiments ($\kappa = 1\%$).

 

**Table 4: One of the key findings from the Wild-Time benchmark [55] is that no existing method consistently outperforms Empirical Risk Minimization (ERM) across all datasets in the Wild-Time benchmark. We show the comparisons between HyperTime and ERM on all datasets of Wild-Time benchmark. All the numbers are the higher the better.**

|  | MIMIC-Readmission | | MIMIC-Mortality | | HuffPost | | Arxiv | | FMoW-Time | | Yearbook | |
|---|---|---|---|---|---|---|---|---|---|---|---|---|
|  | Avg. | Worst | Avg. | Worst | Avg. | Worst | Avg. | Worst | Avg. | Worst | Avg. | Worst |
| ERM | 48.02 | 43.68 | 77.24 | 73.45 | 70.60 | 69.14 | 46.39 | 44.53 | 58.05 | 46.40 | 77.74 | 65.24 |
| HyperTime | **54.81** | **51.44** | **78.26** | **74.52** | **71.68** | **69.72** | **48.48** | **46.52** | **59.17** | **50.02** | **84.58** | **73.91** |

**Table 5: Normalized test results of CFO and HyperTime using chronological and randomly shuffled folds construction methods. We show the average test accuracy and the worst fold accuracy, which are denoted as Test-average and Test-worst.**

|  | Electricity | | | | Vessel Power | | | |
|---|---|---|---|---|---|---|---|---|
| Method | CFO | HyperTime | CFO | HyperTime | CFO | HyperTime | CFO | HyperTime |
| With Chronology | True | True | False | False | True | True | False | False |
| Test-average | 1.855 | **1.333** | 3.189 | 3.826 | 1.378 | **1.008** | 3.272 | 3.068 |
| Test-worst | **1.104** | 1.174 | 3.065 | 3.206 | 1.689 | **1.027** | 3.480 | 3.084 |

**Table 6: Test time results regarding average test accuracy, the worst fold accuracy, and the number of winning folds for a state-of-the-art robust training method LISA [56], our method HyperTime, and the methods adding LISA to CFO and HyperTime respectively.**

|  | LISA | CFO+LISA | HyperTime | HyperTime+LISA |
|---|---|---|---|---|
| Test-average | 83.45 | 84.19 | 84.58 | **85.11** |
| Test-worst | 70.74 | 65.77 | **73.91** | 71.90 |
| WN | 0 | 0 | 3 | **6** |

As shown in Figure 5, the optimization objective formulation in our method is obviously better. There are three takeaways: (1) HyperTime is obviously better than CFO_Worst indicating that both two optimization objectives (average and worst fold performance) should be considered in our method. (2) HyperTime consistently outperforms CFO_WeightedCombine indicates that the importance of formulating the optimization of these two objectives as a lexicographic optimization problem. (3) HyperTime consistently outperforms HyperTime_Reverse indicating that the average validation loss shall be considered an objective of a higher priority compared with the worst-case validation loss.

Additionally, we also conduct additional experiments to compare HyperTime with CFO_WeightedCombine using different weight settings in Appendix D, and we still observe that HyperTime outperforms CFO_WeightedCombine, which further demonstrates the importance of formulating the optimization of these two objectives as a lexicographic optimization problem.

*5.3.2 Compatibility with Robust Training.* Since our method is a generic hyperparameter optimization solution, it is agnostic to the specific learning method as long as there are important hyperparameters to tune. In this subsection, we show the compatibility of our method with robust training methods, which shows its advantage in further boosting the robustness of the whole machine-learning pipeline.

We perform evaluations on the yearbook dataset by adding robust optimization method LISA [56] to HPO which achieves the best performance in Wild-Time [55]. Specifically, we reuse the LISA implementation from Wild-Time and use our algorithm to tune its hyperparameters, including both the architecture hyperparameters and non-architecture hyperparameters. The detailed search space is the same with Section 5.2.2 as shown in Appendix C. Table 6 shows the final overall results and we also include the test results of different folds for each method in Appendix D. We have the observations below:

(1) Combining HyperTime with LISA achieves better average performance compared with using either of them. (2) Combining HyperTime with LISA has more winning numbers compared with all other methods. (3) Combining HyperTime with LISA improves the worst fold performance over LISA, but degrades the worst fold performance compared with HyperTime alone.

In summary, observations (1) and (2) demonstrate that the combination of HyperTime and other non-HPO temporal distribution shift solutions further boost the model performance compared with using either of them. Observation (3) shows one disadvantage of this combination, and it is worth investigating the reason and the method for mitigating it in future work.

## 6 CONCLUSION

In this work, we propose a new method to combat temporal distribution shifts named HyperTime. HyperTime approaches this problem by performing multi-objective hyperparameter tuning with a lexicographic preference across different objectives, on a set of chronologically constructed validation sets. We evaluate HyperTime across multiple datasets and learners, which verify its strong empirical performance even compared with the state-of-the-art robust training methods. Moreover, we also perform experiments to provide a better understanding of the important contributing factors in our method and demonstrate that HyperTime is agnostic of learning methods, and combining it with other non-HPO robust learning methods could further boost the performance.

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
