# OpenReview forum: "HyperTime: Hyperparameter Optimization for Combating Temporal Distribution Shifts"
_acmmm.org/ACMMM/2024/Conference — MM2024 Poster_

### Official Review · Reviewer_xULz · 2024-05-03

**Rating:** 4
**Confidence:** 2

**Summary:**

This paper proposes a method for optimizing hyperparameters to enhance the robustness of machine learning models against temporal distribution shifts in data. The approach, named HyperTime, integrates concepts from robust optimization to prioritize hyperparameters that yield stable performance over time, particularly when future data distributions may differ from those seen during training.

**Strengths:**

1. This paper evaluates the model by considering both average-case and worst-case validation performance, providing a comprehensive view of its robustness across different scenarios.
2. The paper offers an extensive analysis of cross-validation and holdout strategies, utilizing a wide array of datasets. It includes both test-average and test-worst as evaluation criteria. Nonetheless, it remains unclear if these methods adequately mitigate the impacts of temporal distribution shifts.

**Limitations:**

1. Lexicographic optimization might be difficult in the sense that in general it's easy to optimise the first component, but optimising the second component without hurting the first one is diffuclt. Any theoretical justification for your algorithm to achieve this optimization goal or this is just a motivation for the algorithm?
2. In Theorem 2, the assumption on $\kappa$ requires the knowledge about the optimal $L_{avg}(c^*_{k^*})$. It seems in practice this assumption cannot be verified effectively. How do you tell whether your theorem conclusion holds or not.
3. This paper motivates the result by using temporal distribution shifts. However, it does not seem that this optimization method can online learning the temporal distribution shift.

**Suitability:**

2

---

### Official Review · Reviewer_LcUX · 2024-05-29

**Rating:** 4
**Confidence:** 2

**Summary:**

This paper focuses on hyperparameter optimization under distribution shifts and presents a new method named HyperTime, inspired by the "worst-case-oriented" philosophy. Unlike traditional methods that solely consider the average loss across different validation sets, HyperTime incorporates the worst loss as a secondary objective, utilizing lexicographic hyperparameter optimization. Both theoretical analysis and experimental results effectively demonstrate the method's robustness and effectiveness.

**Strengths:**

1. The use of Lexicographic Hyperparameter Optimization allows for the thorough optimization of $L_{avg}$ while also considering $L_{worst}$.
2. This paper provides a theoretical analysis and discusses the selection of $\kappa$.
3. The notation is clear, the logic is rigorous, and the intuitive insights, algorithm, theoretical analysis, experimental validation, and ablation study are all well-connected.

**Limitations:**

1. There is a typo in Line 106.
2. HyperTime adds $L_{worst}$ as an additional loss. Compared to the baseline, does this introduce extra computational overhead? If so, how much additional overhead is there?
3. In the "Relevance to Conference" section, the authors claim that "there exist severe temporal distribution shift problems in multimodal data." However, the experiments were all conducted on unimodal data.

**Suitability:**

2

---

### Official Review · Reviewer_AmkT · 2024-05-31

**Rating:** 4
**Confidence:** 2

**Summary:**

Summary:
This work studies the Hyperparameter Optimization problem to solve the distribution shift problem during temporal sequence learning. Specifically, based on the observation that it is possible to achieve temporally robust performance via hyperparameter optimization. To find such hyperparameters, the authors propose HyperTime method which leverages the “worst-case optimization” to achieve robust learning performance. By comparing worst-case optimization to average optimization, the effectiveness of HyperTime is carefully validated. Moreover, rigorous theoretical analysis and various experimental evaluations are conducted to justify the proposed method.

**Strengths:**

Strengths:
- This paper is well-written and easy to follow
- The motivation is clear, there are various evaluations proposed to carefully justify the proposed method.

**Limitations:**

Weaknesses:
- One of the key drawbacks of worst-case optimization is that it encourages overfitting on the worst-case distribution. If the proposed method employs DRO as one of the objectives, I am curious about whether this design leads to overfitting. Could the authors justify that perspective?
- The motivation for employing Lexicographic priority is uncertain. Why such a priority is important for solving this problem? Is it possible that other priority measures could be effective than the current choice? Or is it possible that simply combining average term and worst-case term together can still achieve similar performance?

**Suitability:**

2

---

### Official Review · Reviewer_LUNa · 2024-06-04

**Rating:** 4
**Confidence:** 3

**Summary:**

This work modeled machine learning undering temporally shifting distributions as lexicographic hyperparameter optimization problems. The authors construct the lexicographic objective with standard average validation loss as the first priority and worst-case validation loss as the second priority. With validation sets carefully constructed and the direct apply of lexicographic optimization to hyperparameter optimization (HPO), the resulting machine learning model possesses better robustness to temporal distribution shifts. The authors empirically verified the robustness.

**Strengths:**

+ The modeling of temporally robust machine learning optimization as lexicographic hyperparameter optimization is novel and interesting.
+ The empirical results seemed to be very positive.

**Limitations:**

- Techincal contribution is weak as this work is a direct application of existing optimization method to a new scenario.
- The theoretical analysis did not provide much guidance on how to apply the proposed method. Its implicaiton, the role of $\kappa$, is not empirically verified either. Since $\kappa$ itself is a hyperparameter of the optimization algorithm, the authors did not provide any numerical evidence as to how much impact it has on the final performance and how hard it is to tune this parameter.
- The $I$ definition in line 288 lacks some context, especially when the $|\cdot|$ operator is not defined for ordered lists either. It confused me at first glance.

**Suitability:**

3

---

### Meta-Review · Area_Chair_r3wd · 2024-06-27

**Recommendation:** Accept (Poster)
**Confidence:** 4

**Metareview:**

This work formulates machine learning under temporally shifting distributions as lexicographic hyperparameter optimization problems.  To solve it, the authors propose the HyperTime method to achieve robust learning performance. The effectiveness of HyperTime is carefully validated.

All reviewers agree that the paper is a solid contribution with clear motivation, a sound approach, and extensive evaluation.

I recommend acceptance.